# Portal Vein Embolization: Rationale, Techniques, and Outcomes to Maximize Remnant Liver Hypertrophy with a Focus on Contemporary Strategies

**DOI:** 10.3390/life13020279

**Published:** 2023-01-19

**Authors:** Jonathan Charles, Nariman Nezami, Mohammad Loya, Samuel Shube, Cliff Davis, Glenn Hoots, Jamil Shaikh

**Affiliations:** 1Morsani College of Medicine, University of South Florida, 560 Channelside Drive, Tampa, FL 33602, USA; 2Department of Diagnostic Radiology and Nuclear Medicine, Division of Vascular and Interventional Radiology, University of Maryland School of Medicine, 655 W Baltimore St. S, Baltimore, MD 21201, USA; 3Experimental Therapeutics Program, University of Maryland Marlene and Stewart Greenebaum Comprehensive Cancer Center, 22 S Greene St., Baltimore, MD 21201, USA; 4Mid-Atlantic Permanente Medical Group, Kaiser Permanente, 2101 E Jefferson St., Rockville, MD 20852, USA; 5Department of Radiology, Tampa General Hospital, University of South Florida Health, 1 Tampa General Cir., Tampa, FL 33606, USA

**Keywords:** portal vein embolization, future liver remnant, liver hypertrophy, hepatic vein deprivation, liver functional reserve

## Abstract

Hepatectomy remains the gold standard for curative therapy for patients with limited primary or metastatic hepatic tumors as it offers the best survival rates. In recent years, the indication for partial hepatectomy has evolved away from what will be removed from the patient to the volume and function of the future liver remnant (FLR), i.e., what will remain. With this regard, liver regeneration strategies have become paramount in transforming patients who previously had poor prognoses into ones who, after major hepatic resection with negative margins, have had their risk of post-hepatectomy liver failure minimized. Preoperative portal vein embolization (PVE) via the purposeful occlusion of select portal vein branches to promote contralateral hepatic lobar hypertrophy has become the accepted standard for liver regeneration. Advances in embolic materials, selection of treatment approaches, and PVE with hepatic venous deprivation or concurrent transcatheter arterial embolization/radioembolization are all active areas of research. To date, the optimal combination of embolic material to maximize FLR growth is not yet known. Knowledge of hepatic segmentation and portal venous anatomy is essential before performing PVE. In addition, the indications for PVE, the methods for assessing hepatic lobar hypertrophy, and the possible complications of PVE need to be fully understood before undertaking the procedure. The goal of this article is to discuss the rationale, indications, techniques, and outcomes of PVE before major hepatectomy.

## 1. Introduction

### 1.1. Hepatectomy

Rates of both primary and secondary liver cancer has been increasing in incidence over the past few decades [1], with hepatocellular carcinoma (HCC) being the fourth most common cause of cancer-related mortalities worldwide [2]. Liver transplantation and surgical resection are the only two therapies that offer long term survival, and because of the strict criteria for transplantation in addition to the finite number of available organs, resection, when possible, remains the mainstay treatment in both HCC and confined metastatic disease [3,4]. Additionally, patients must meet certain criteria in order to be amenable to surgical resection [5]. Large-volume tumoral liver resections carry the substantial risk of post-hepatectomy liver failure (PHLF) due to the inability of the residual hepatic tissue to handle the workload of the previous whole organ.

### 1.2. Future Residual Liver Volume

The percentage of liver that remains post-resection, known as the future liver remnant (FLR), is an independent and reliable predictor of post-resection hepatic dysfunction [5,6]. The expected absolute FLR volume alone is not enough to predict a good prognosis for patients post-resection. Patients with larger body mass indices (BMI) and body surface areas (BSA) will require a larger FLR to achieve appropriate compensation to avoid hepatic insufficiency. Therefore, the concept of the standardized FLR (sFLR) emerged to standardize the FLR relative to a patient’s size [7]. The sFLR, expressed as a percentage of the patient’s liver volume, is determined by taking the FLR as a ratio of a patient’s total functional liver volume (TLV). Multiple studies have shown that post-resection complications are significantly reduced in patients with an sFLR > 20% in normal livers [8]. Interventional oncologists offer a unique procedure to achieve this, and thereby expand resection indications, limiting dropouts from curative treatment and improving management algorithms. Much of the literature continues to use FLR as a measurable outcome, and we will continue to report FLR for the purposes of this review.

A second associated concept referred to as kinetic growth rate (KGR) has been shown to be a promising factor in determining mortality from hepatic insufficiency [9]. KGR is calculated by degree of hypertrophy at first post-PVE volume assessment (%) ÷ time elapsed since PVE (weeks) at the first post-PVE volume assessment. A recent study demonstrated KGR > 2.0% growth per week was associated with zero cases of hepatic insufficiency post-resection. The authors concluded KGR may be a better predictor of postoperative morbidity and mortality after liver resection than conventional volume parameters.

### 1.3. Portal Vein Ligation

The role of portal vein ligation (PVL) in inducing hypertrophy of the liver has been clearly demonstrated in an experimental study studying liver hypertrophy in rat models [10]. However, another report has suggested that PVL was significantly less efficient than portal vein embolization (PVE) in inducing hypertrophy of the left lateral segments [11]. Pandanaboyana et al. took this a step further and published a meta-analysis comparing PVL with PVE for elective liver resection. A total of 218 patients were included, and they found no significant difference in FLR hypertrophy between the two groups (PVE: 39%, PVL: 27%, mean difference 6.04, 95% confidence interval −0.23, 12.32, *p* = 0.06). This procedure achieves liver hypertrophy by surgically ligating the desired extrahepatic branch of the portal vein, thereby re-directing the entire portal vascular flow into the FLR. A common topic of discussion is the relative efficacy of PVL versus PVE for inducing FLR hypertrophy [12]. Capussotti et al. demonstrated that PVL is just as efficacious as PVE in inducing FLR hypertrophy [13]. A larger systematic review by Vyas et al. found that PVL induced a mean FLR hypertrophy rate of 64.65%, with a receptibility rate of 63.68% versus a PVE hypertrophy and resectability rate of 39.75% and 76.88%, respectively [14]. 

### 1.4. Portal Vein Embolization

Since its inception almost 30 years ago, PVE has now become the standard of care treatment to prevent post-hepatectomy liver failure (PHLF). During PVE, embolic material is administered intra-vascularly into select intrahepatic portal veins to decrease the portal vascular flow to the targeted liver segments with tumoral involvement. Occlusion of the portal veins ultimately deprives the embolized segments of liver of the blood flow required to sustain growth, while subsequently inducing a physiologic response to hypertrophy in the non-embolized liver segments. PVE leverages the unique dual vascular supply of the liver, with the liver parenchyma being fed by both hepatic arteries and from the portal venous system which reduces the risk of infarction. 

To date, multiple studies have proved PVE to be safe and effective in down-staging tumoral liver tissue, optimizing future liver remnant (FLR) and ultimately increasing the number of patients eligible for major liver resection [15,16,17].

However, there is also published evidence that points towards PVE having controversial results in overall survival (OS) rates, as well as on disease-free survival (DFS) rates. Giglio et al. performed a systematic review analyzing the oncological outcomes of patients who underwent major liver resection following PVE, and they reported on postoperative hepatic recurrence (PHR) and 3-year and 5-year OS rates between patients who received PVE and those who did not [18]. They found that no difference in PHR (*p* = 0.41), 3-year OS (*p* = 0.22), and 5-year OS (*p* = 0.82) was noted between the two groups. These results were consistent with other published literature [19,20,21]. In reference to DFS, Ardito et al. analyzed liver-specific DFS curves for two groups, one receiving PVE and one not [21]. They found that at 5 years, no significant difference in DFS was observed between both groups (*p* = 0.572), and even found that patients in the PVE group had experienced recurrence of colorectal cancer liver metastasis earlier than the non-PVE group, although the rate of overall intrahepatic recurrence was not significantly different between the two groups (*p* = 0.749). 

This is to say that PVE is a promising procedure that carries much benefit, but is not without its complications and drawbacks, and operators need to be well aware of both aspects of the procedure to effectively care for their patients.

## 2. Indications and Contraindications

The primary indication for pre-operative PVE is centered on the inability of the pre-embolization FLR to support whole liver function post-resection. With this in mind, several factors, such as the patient’s baseline hepatic function, the size of the liver portion to be resected, complexity of the planned resection (i.e., extended right hepatectomy), age, and co-morbidities, are taken into account when determining patient eligibility. The adequacy of the FLR should be assessed using both the remnant volume and the remnant liver function. The FLR volume can be assessed with the FLR ratio, which can be calculated as FLR volume/TLV.

In patients with an otherwise normal underlying liver, PVE is indicated if there is an FLR < 20%, or an FLR to body weight ratio (FLR–BWR) ratio of < 0.5%, according to the Truant criterion [22]. In patients who have underlying liver dysfunction, including exposure to hepatotoxic chemotherapy or hepatic steatosis, PVE is considered for patients with an FLR < 30% or an FLR–BWR of <0.8% [23,24,25,26]. It should be noted here for transparency that there is literature suggesting that systemic chemotherapy did not impair liver hypertrophy in the setting of PVE [27,28,29]. Therefore, it remains up to the operator and the patients through their shared decision making whether to move forward with PVE. There remains a third demographic for which a different FLR cut-off applies for PVE consideration. Patients with Childs-Pugh class A cirrhosis require yet a higher expected FLR < 40% and an FLR -BWR ratio < 1.4% [30,31]. However, it needs to be noted here that in addition to Child-Pugh status, indocyanine green retention at 15 min (ICG-R15) is also often incorporated into the treatment algorithm, thereby changing the FLR percentage cutoff. MD Anderson Cancer Center utilizes the 40% FLR cutoff if the patient has Child A cirrhosis with a normal ICG-R15 (<10%), but will require a 50% FLR if the ICG-R15 is between 10–20% [32].

Contraindications to PVE are severe portal hypertension, uncontrollable intrahepatic portal-to-hepatic vein shunts, tumor thrombus in the portal vein, and occlusion of the portal vein in the FLR. Patients with extensive distant metastatic disease or periportal lymphadenopathy cannot undergo resection, and therefore are not candidates for PVE.

## 3. Methods for Portal Vein Access

PVE can be performed through two approaches: open surgical (transileocolic) and percutaneously via a transhepatic route with ultrasound guidance. In recent years, the open surgical methods to achieve liver hypertrophy have fallen out of favor given the higher risks with open surgery and, more importantly, that open techniques produce significantly lower FLR hypertrophy [33]. Percutaneous techniques for PVE utilize both ipsilateral and contralateral transhepatic accesses. Contralateral access (that is access via the FLR liver parenchyma) allows for ease of catheter manipulation and deployment of embolic material. The ipsilateral approach however is preferred as it gains access to the portal system through tumoral liver segments and preserves the FLR segment in case of complications. Percutaneous trans-splenic access has also been proposed with a reported technical success rate of 88.9% and a major complication rate of 3.8%, with the primary complication including splenic vein dissection [34]. 

It should be noted however, that the choice of approach is ultimately at the discretion of the operator and can rely on multiple factors such as the tumor burden, the extent of the embolization, and the comfortability of the operator. Hepatic contrast-enhanced computed tomography (CT) scans with 5.0 mm or less slice thickness are obtained prior to PVE, allowing for accurate visualization of the tumoral and non-tumoral liver segments.

### 3.1. Ipsilateral Approach

Directly before the procedure, patients receive a single dose of intravenous ceftriaxone 1 g (Roche, Nutley, NJ, USA). Patients are typically placed under conscious sedation or general anesthesia as per hospital protocol. Under ultrasound guidance, a distal portal branch is accessed through the tumoral liver tissue via a 21 G Chiba needle (Merit Medical, South Jordan, UT, USA). Anterior branches of the right portal vein tend to be targeted here because they have been associated with lower complication rates [16]. Once an adequate vein has been accessed, contrast agent is injected to confirm positioning and a microwire is passed centrally. Use of a non-vascular access kit such as Neff Set or Accustik is used to upsize the microwire to an 0.035 system, through which an 0.035 guidewire is passed centrally into the main portal vein. Using standard Seldinger technique the access site is upsized to a 5 or 6 F vascular sheath. It should be noted that although the ipsilateral approach spares puncture through the FLR, it can be problematic in patients with large tumor burden, as access through the tumor increases the risk of peritoneal seeding [16].

### 3.2. Contralateral Approach

Similar to the ipsilateral approach, a distal portal branch of segment 2 or 3 is accessed under ultrasound guidance using a 21 G Chiba needle. Typically, because of the angle of the portal vein confluence, a segment 3 branch is preferred so the route to the right portal system has less tortuosity. It should be noted that the contralateral approach has the advantage of easier catheter manipulation to and in the right hepatic segments given the angle of access; however, it requires puncture through the non-tumoral tissue which opens the possibility for damage to the FLR and the associated complications of vascular access [16].

## 4. Embolization Techniques

After access into the portal vein, a catheter is passed centrally into the main portal vein. Digital subtraction portography is obtained via a pigtail catheter to evaluate the anatomy and assess the target veins. In typical right hepatectomy cases, the right portal branches of the anterior and posterior division are targeted for embolization. In cases of extended right hepatectomy, segment 4 A and 4 B branches may also be targeted for embolization. Embolization of segment 4 does raise concern for possible inadvertent embolization to the FLR, however recent studies have shown improved growth of segments 2 and 3 as well as higher KGRs after embolizing segment 4 [35]. Ito et al. demonstrated hypertrophy of segments 2 and 3 in patients who underwent right PVE with and without segment 4, of 52.4% versus 32.2%, and KGRs of 3.1% per week versus 2.0% per week, respectively [36]. 

However, a subject of debate within the current practice of PVE today is focused on the choice of embolic agent. The goal of PVE is to achieve complete portal occlusion of the targeted liver segments while ensuring targeting of all disease segments to maximize hypertrophy of the FLR and prevent hypertrophy of the tumoral segments. The ideal embolic should be tolerated by the patient, cause a complete occlusion without future recanalization, be cost-effective, and be simple to administer. Multiple methods have been described in the literature, but none have been shown to have superiority. 

Current research in PVE techniques is evaluating which embolization technique achieves FLR growth the fastest and thereby allows for earlier surgery. Embolization can be achieved using polyvinyl alcohol (PVA) particles, microspheres, gelfoam, N-butyl cyanoacrylate glue (nBCA) or sodium tetradecyl sulfate (STS) foam with or without combination coils/plugs. To date, only one recent, randomized prospective study (BEST-FLR Trial) is available, demonstrating superiority of nBCA glue over particles. It showed greater and faster liver growth (FLR increased 57% vs. 37%, and KGR 4% per week vs. 3% per week, respectively) [37].

All successful PVE techniques achieve occlusion of the target segments and include both distal occlusion of the portal veins to constrain the development of intrahepatic collaterals and proximal occlusion to prevent venous inflow. 

### 4.1. Embolic Materials

#### 4.1.1. PVA/Microparticles and Coils

PVA particles/microparticles range in size from 150–1000 um and are widely commercially available for use. Because of their varying sizes, they are an excellent option for distal embolization. Typically, operators will use these particles in combination with coils or plugs for proximal embolization to provide complete embolization of the target portal veins (Figure 1).

In study by Camelo et al., 63 patients who received preoperative PVE with PVA particles and coils demonstrated an FLR increase from a mean value of 484 mL ± 242 to 654 mL ± 287 (*p* < 0.001), a mean percentage increase of 40% [38]. It should be noted that this study found a negative relation between the FLR volume before PVE and the FLR volume increase induced by PVE (R = −0.46, *p* < 0.001). Two out of the 64 patients (3.1%) suffered major adverse events. Ultimately, 44 patients underwent successful surgical resection (68.8%), with only one with postoperative hepatic insufficiency.

#### 4.1.2. N-Butyl Cyanoacrylate Glue + Lipiodol

nBCA glue is a liquid embolic agent that has been shown to be effective for PVE and is commercially available in small aliquots (typically 1 mL vials). The embolic agent polymerizes when it comes into contact with an ionic agent and forms a permanent bond to adjacent structures. Typically, the agent is diluted with lipiodol which slows polymerization and allows the embolic agent to be radio-opaque on fluoroscopy (Figure 2).

The ideal ratio of nBCA to lipiodol is not established but depends on how distal embolization needs to be achieved; ratios between 1:5 and 1:9 are used and administered in small 0.5 to 1.0 mL aliquots until stasis is achieved. When the portal embolization is complete, the punctured branch can be clotted off with either administration of additional nBCA and lipiodol or with manual compression while removing the access sheath. nBCA glue results in significant peripheral inflammation and produces effective portal occlusion. 

Mukund et al. assessed PVE with nBCA glue and lipiodol in 28 patients [39]. The mean absolute FLR volume increased from 371 mL ± 87 to 567 mL ± 142, with a mean percentage increase of 52% ± 32 (*p* <0.0001). A total of 18 of the 28 patients (64.28%) underwent successful surgical resection 4–8 weeks post-PVE. Of the patients who underwent surgical resection, only one developed transient post-operative hepatic failure on post-operative day 5 but ultimately recovered on post-operative day 10.

#### 4.1.3. Sodium Tetradecyl Sulfate Foam

Another liquid embolic/sclerosing agent which has been gaining favor recently has been sodium tetradecyl sulfate (STS) foam. Its low cost and ease of use has become a factor in its adoption. The liquid sclerosant is turned into a foam by the mixture of air and oil-based contrast agent (lipiodol), which allows full contact with the vascular endothelium when injected into the venous system. Theoretically, foam allows for higher surface area contact and makes it a better embolic agent than liquid agents. The agent itself results in inflammation and subsequent thrombosis/sclerosis upon contact with vascular endothelium [40,41] (Figure 3).

In a single-center retrospective review, Fischman et al. described successful PVE and FLR growth in 35 patients with STS foam [42]. Adequate FLR hypertrophy was achieved in 31 of the 35 patients (88.6%) at 30 days. The mean percentage increase in FLR was observed at 11.9% ± 10.2. Importantly, the authors described absolute and percentage FLR increases that did not significantly differ based on type of underlying liver function (healthy, steatosis, chemotherapy, cirrhosis). Fever and post-embolization syndrome were reported in 10 of the 35 patients (28.6%) a much larger cohort compared to other embolic agents. A total of 27 of the 35 patients (77.1%) underwent surgical resection, with no reported 30-day fatalities.

#### 4.1.4. Absolute Ethanol

Absolute ethanol can be used in PVE as a reliable embolic due to its strong coagulative effect and the low risk that it carries for vascular recanalization. Absolute ethanol causes tissue fixation, sludging of blood cells, and protein denaturation and coagulation when administered intravascularly. Yamakado et al. described significant hepatic necrosis in post-ethanol embolization via histopathological analysis. The group furthermore outlined the presence of a dose-dependent relationship between the degree of parenchymal damage and the amount of ethanol injected [43]. Additionally, absolute ethanol has the advantages of being readily available, very cost-effective, and easy to administer given its low viscosity. 

Sofue et al. studied the outcomes of PVE in 83 patients with absolute ethanol (99.5% ethanol; Fuso Pharmaceutical Industries, Osaka, Japan) [44]. The mean FLR increased after PVE from 366 mL to 513 mL, a 39.9% increase (*p* < 0.001). Four of the 83 patients experienced post-embolization adverse events, but no complications precluded hepatic resection. A total of 69 of the 83 (83%) patients underwent hepatic resection at a median of 25 days post-PVE with no post-operative mortality noted.

#### 4.1.5. Powdered Gelfoam

The use of gelfoam powder takes a slightly different approach towards PVE in relation to the duration of the therapy post-embolization. Whereas the previously mentioned embolics all function as permanent embolization techniques, the use of powdered gelfoam for PVE is a temporary technique. The use of permanent embolic agents is typically associated with several drawbacks including periportal inflammatory fibrosis of the perivascular connective tissue which can lead to a difficulty in hilar dissection as well as post-embolization pain. Additionally, there is theoretically an increased risk of extension of portal thrombosis in patients with reduced hepatopetal portal flow along with partial liver parenchymal necrosis when absolute ethanol is the embolic of choice. In contradistinction, use of gelfoam during PVE allows for revascularization of the portal vascular bed if the liver is not ultimately resected. Secondly, any unwarranted migration of embolic agent into non-desired portal branches would not preclude liver regeneration [45]. 

An article by Lainas et al. challenges the established concept of permanent and total occlusion of the portal vein to induce efficient FLR hypertrophy [46]. They assessed the effect of reversible PVE through powdered absorbable gelfoam (Curaspon, Curamedical, Zwaneburg, The Netherlands) on liver regeneration using nine monkeys. After the initial embolization, complete portal vein recanalization occurred on post-embolization days 12–16. It induced a significant increase in FLR/TELV volume of 16.4%, from 38.4% to 54.8%, measured one month post-embolization.

#### 4.1.6. Post-Portal Vein Embolization Operative Complications

No procedure is without its complications. In a large review of 1179 patients who received PVE, a procedure-related mortality of 0.1% and a major complication rate of 0.4% was determined [47]. In this cohort, major complications included infection, non-target embolization, vascular injury, portal and/or mesenteric venous thrombosis, portal hypertension, and biloma. Minor complications included fever (36.9%), transaminase elevations (34.8%), abdominal pain (22.9%), nausea/vomiting (2.0%), and ileus (1.2%) and were generally self-limited. Another potential major complication of PVE is growth of the tumoral liver tissue itself. Some reports have shown accelerated tumor growth in the liver after PVE; however, this can be mitigated when all of the tumor-bearing areas are carefully mapped and embolized [48]. Other complications are not necessarily specific for transhepatic PVE but apply to any percutaneous intervention. These include but are not limited to bleeding, infection, pseudoaneurysm, and arteriovenous or arterio-biliary fistula formation.

A summary of the previous five embolics is detailed in Table 1 below.

## 5. Future Directions/Goals to Enhance Future Liver Remnant Growth after Portal Vein Embolization

The field of FLR maximization and regeneration is an actively evolving one, that incorporates novel methods in order to provide the best therapy that is supported by significant evidence. PVE with adjunctive therapy has been researched and, in some cases, utilized in practice to further optimize the regenerative capacity of the embolization procedure to achieve a faster or more effective FLR rate of growth.

### 5.1. Dietary Supplementation to Maximize Future Liver Remnant Volume Increase

Generally, malnutrition is frequently associated with liver disease, and therefore, proper nutritional support might be necessary to improve the outcomes of liver disease treatment [57]. Branched-chain amino acids (BCAA), those amino acids having an aliphatic side-chain with a branch, comprise three essential amino acids, namely leucine, isoleucine, and valine. BCAA supplementation improves cellular metabolism, amino acid transport, and protein turnover. Moreover, BCAAs activate mammalian target of rapamycin (mTOR) signaling, stimulating the synthesis of glycogen and albumin, cell growth and proliferation, insulin resistance, and the phosphoinositide-3-kinase-protein kinase B (PI3K-Akt) signaling pathway [58]. Therefore, they are thought to promote liver regeneration and accelerate liver recovery after treatment-related damage.

A randomized clinical study enrolling 28 patients sought to compare the effects of BCAA supplementation (LIVACT^®^, Ajinomoto, Tokyo, Japan)-initiated pre-PVE and carried through 6 weeks after liver resection versus those who did not receive any supplementation [59]. The authors measured FLR through liver uptake on technetium-99 m galactosyl human serum albumin scintigraphy. The authors reported that the supplementation group demonstrated a liver uptake increase of 266.7% vs. an increase of 77.6% in the PVE alone group (*p* = 0.04).

### 5.2. Associating Liver Partition and Portal Vein Ligation for Staged Hepatectomy

Associating liver partition and portal vein ligation for staged hepatectomy (ALPPS) involves a surgical procedure consisting of right portal ligation and in situ splitting of the liver parenchyma on the right side of the umbilical portion of the portal vein. This procedure is staged in two components: stage 1 and stage 2. Stage 1 ALPPS is the procedure in which the portal vein ligation and in situ surgical splitting of the liver parenchyma occurs, and stage 2 is the surgical resection of the desired liver segments. This is a procedure that has been reported to have been performed on patients with insufficient FLR hypertrophy following PVE, with the sentinel report of this procedure reporting an FLR hypertrophy of 63% within a median of 8 days [60]. A few studies have shown that mean FLR hypertrophy after sequential PVE + ALPPS ranges from 41.7 to 88% [61,62,63], leading to a consensus statement from an international ALPPS expert meeting recommending that adjunctive ALPPS can be considered for post-PVE patients with insufficient hypertrophy. From these data, it can be suggested that ALPPS provides a significantly greater FLR hypertrophy rate when compared to PVE alone. However, there are significant concerns about the ALPPS procedure as it relates to morbidity and mortality. The reported 90-day mortality after ALPPS is 15%, compared to 6% in PVE, and the odds ratio for perioperative death was 2.7-fold higher in the ALPPS patients [64]. Based on this, it was concluded in a report that PVE and interval resection remain the standard of care for patients with small pre-embolization FLRs [65]. However, for the predominant indication for ALPPS being colorectal-liver metastasis, the perioperative mortality and morbidity rates are slightly improved. Schnitzbauer et al. demonstrated an initial perioperative mortality of 8%, with a possible selection bias given that ALPPS might have been performed in patients with more extensive hepatic tumor burden [66]. 

A novel approach towards the ALPPS procedure was described by Santibañes et al. in 2016, dubbed the “MINI-ALPPS” procedure [67]. They proposed incorporating a combination of partial parenchymal transection with intraoperative PVE with minimum liver mobilization. This procedure essentially avoids a large parenchymal division as well as hilar plate or hilum dissection, thereby minimizing liver mobilization. Moreover, the traditional PVL is swapped with PVE. They reported a mean FLR hypertrophy rate of 62.6% in a median of 11 days in four patients. Collectively, none of the patients required a blood transfusion during the first stage, and during the second stage, three patients received a median of two blood units, much less than the reported blood loss in the traditional ALPPS procedure. Pekolj et al. went further and successfully performed the MINI-ALPPS procedure laparoscopically on one patient [68].

### 5.3. Percutaneous Intrahepatic Split by Combing Portal Vein Embolization and Ablation

As mentioned in the section above, the main drawback of the ALPPS procedure is the extremely high morbidity and mortality associated with the first-stage of the procedure. Consequently, modifications to the ALPPS procedure were quickly proposed after its conception. The aim of the newer techniques is to reduce the trauma associated with stage 1 ALPPS by either avoiding hilum dissection or making the in situ split of the liver less traumatic. PVE + percutaneous intrahepatic split by ablation (PISA) avoids a formal surgical operation with long-lasting general anesthesia, as well as limited consequence on patient safety in the event of a failed progression to stage 2 ALPPS. Lunardi et al. developed a method for PISA in order to reduce trauma associated with the ALPPS procedure [69]. Six patients were enrolled who received either PVE alone, or PVE + PISA. The PISA technique described in this work utilizes a microwave-ablation system (14 G-20 cm Micro ThermX, Perseon, Salt Lake City, UT, USA) to ablate the liver parenchyma along the main portal fissure, following the future surgical plane. No procedure-related complications were recorded after the PVE and PISA procedures. The patients who received PVE + PISA saw an average FLR increase of 82.9% 10 days post-PISA (3 weeks post-PVE). After surgical resection, patients were discharged on post-operative days 14–19 and had no 90-day mortality.

### 5.4. Liver Venous Deprivation: Simultaneous Portal Vein and Hepatic Vein Embolization 

Recently, combined simultaneous embolization of portal and hepatic veins has been described and the first studies show it to be a safe and feasible technique with faster growth rates than PVE alone [70,71,72]. This simultaneous PVE and hepatic vein embolization (HVE), commonly referred to as liver venous deprivation (LVD), involves embolizing both the desired portal vein branch and ipsilateral hepatic vein branch either during the same or in separate procedures. Because of insufficient FLR hypertrophy following a right PVE, patients can subsequently undergo a right HVE. In these cases, a vascular plug is placed in the central right hepatic vein, and coils embolize the branches. 

A pilot study enrolling 7 patients reported a mean FLR percentage volume increase from 28.2% to 40.9% at 23 days following LVD [71]. In a follow-up study of 10 patients, FLR percentage volume was increased up to 53.4% as early as 7 days following LVD [72]. Recent studies have further demonstrated the efficacy of LVD, even when compared to PVE alone or to ALPPS [73,74,75]. 

A subject of debate surrounding LVD, like that in PVE, is that of oncological outcomes of the procedure. Khayat et al. retrospectively reviewed 17 patients and reported on their 1-year OS, their 3-year OS, and median DFS [76]. They found that the 1-year and 3-year OS post-LVD was 87% and 60.3% respectively, and the median DFS was 6 months. These results showed similar outcomes compared to those observed post-PVE.

### 5.5. Transarterial Chemoembolization and Portal Vein Embolization

There are instances in patients where HCC and liver cirrhosis can form arterioportal shunts, thereby reducing the efficacy of PVE in stimulating hypertrophy as blood flow can be siphoned from the arterial system. It has been shown that transarterial chemo-embolization (TACE) followed by PVE after 1–6 weeks increases the FLR hypertrophy rate more than PVE alone. In one study following 36 patients with cirrhosis and HCC, 18 of whom received TACE 3–4 weeks prior to PVE, the mean relative increase in FLR was greater in the TACE + PVE group compared to PVE alone (12% vs. 8%; *p* = 0.022) with a rate of hypertrophy > 10% being seen at higher instances in the combination therapy group [77]. Most notably, hepatic insufficiency resulting in death was not seen in the 17 patients who demonstrated an increase in FLR volume > 10% 

In contradistinction however, the opposite scenario where TACE is performed after PVE has not been found to yield similar outcomes. A study comparing PVE + TACE where TACE was performed after PVE yielded an FLR increase of 1.4-fold compared to a 1.3-fold increase seen in the PVE group alone [78]. Furthermore, rates of atrophy did not significantly differ either, where the embolized right lobe had atrophied down to 0.75-fold of its original volume in the combination group versus 0.81-fold for the PVE alone group. Coupled with this are increased incidences of hepatic infarction and abscess formation in patients who received TACE after PVE [79,80]. Therefore, careful selection and monitoring of patients who receive combination TACE + PVE is paramount as many of these patients have pre-existing liver dysfunction and are more sensitive to changes in hepatic arterial blood flow.

### 5.6. Transarterial Radioembolization and Portal Vein Embolization

Transarterial radioembolization (TARE) consists of the intra-arterial injection of microspheres impregnated with radioactive radioisotope yttrium-90 (Y90). The unstable Y90 radioisotope undergoes a beta decay, and in doing so, emits beta particles that emit direct cytotoxic damage to the tumor tissue. This procedure has gained ground in the treatment of liver malignancies, namely HCC, with reports suggesting the efficacy of TARE for induction of tumor necrosis [81]. Patients with unresectable HCC and an estimated life expectancy of three months are considered eligible to receive TARE. 

Bouazza et al. published a case report of a patient with a large unresectable segment VIII HCC [82]. The patient received 97 mCi of Y90 TARE, followed by right PVE 11 weeks post-TARE. Pre-TARE tumor volume and FLR/TLV percentage were 1548 mL and 15%, respectively. Six weeks post-PVE tumor volume and FLR/TLV percentage were 717 mL and 27%, respectively. These authors suggested that TARE in combination with PVE may be more advantageous than TACE with PVE. They theorized that given the smaller particle sizes used in TARE relative to TACE, TARE may be less harmful since the smaller particle sizes allow for more distal and selective embolization. Additionally, they argue that TARE + PVE is safer given TARE’s ability to be used in patients with portal vein thrombosis, a situation that is contraindicated in TACE. Finally, they propose that TARE may be working synergistically with PVE in inducing FLR hypertrophy through TARE’s atrophy of the irradiated liver parenchyma, thereby further promoting further compensatory regeneration of the FLR segments [81]. 

### 5.7. Stem Cells as an Adjunct to Portal Vein Embolization

Am Esch et al. evaluated the adjunctive role of hematopoietic stem cell infusion into the portal vein during PVE [83]. A comparison of 22 patients who received PVE alone vs. PVE with simultaneous administration of CD133+ bone marrow stem cells demonstrated an absolute gain in FLR volume of 138.66 mL with the addition of stem cells versus an absolute gain of 62.9 mL in the PVE only group (*p* = 0.004). The adjunctive administration of stem cells during PVE is presumed to accelerate liver parenchyma regeneration via paracrine mechanisms (secretion of cytokines and growth factors that stimulate the growth and differentiation of hepatocytes and cholangiocytes). Furst et al. evaluated 13 patients who received PVE, six of whom also received CD133+ bone marrow stem cells [84]. The cohort receiving the combination therapy saw a mean absolute FLR percentage increase of 77.3%, vs. the PVE alone group’s mean absolute FLR percentage increase of 39.1% (*p* = 0.39). Further research is required as the dosing, route of administration, and efficacy in patients with co-morbid conditions such as diabetes mellitus is unknown. Additionally, the possibility of bone marrow stem cell administration contributing to tumor progression has been suggested in animal models [85]. 

A summary of the novel PVE embolics and techniques are detailed in Table 2.

## 6. Conclusions

Preoperative PVE broadens the spectrum of patients who are eligible to receive curative hepatic resections for their liver malignancies, while also demonstrating survival rates equivalent with those observed in patients who received surgical resection alone. The significant differentiation is that the patients who received the preoperative PVE tended to have considerably more severe disease than the patient who received resection only. Moreover, the procedure itself can be tailored in several different ways based on available resources, patient eligibility, and operator preference. This review detailed the two major portal vein access approaches in PVE procedures and showed why one approach may be more suitable than the other in a given circumstance. Further, we delved into the literature and technique surrounding five major embolics commonly used during PVE: PVA particles with coils; NBCA glue with lipiodol; STS foam; absolute ethanol; and powdered gelfoam, and provided respective data about their efficacy in inducing FLR growth, their major and minor complication rates, as well as their ultimate resection rates. The future of PVE is bright and constantly developing, as demonstrated by the litany of research being conducted about potential adjunctive procedures to PVE that can further bolster the rate of successful hepatic surgical resection. Moving forward, we predict PVE will become a common place tool in the armamentarium of the hepatic surgeon and interventional radiologist to curatively resect hepatic malignancies in eligible patients, with application extending into more atypical and complex liver resections. Optimizing the method of embolization and achieving faster growth of the FLR can result in faster time to resection and definitive management for these otherwise difficult oncologic patients.

## Figures and Tables

**Figure 1 life-13-00279-f001:**
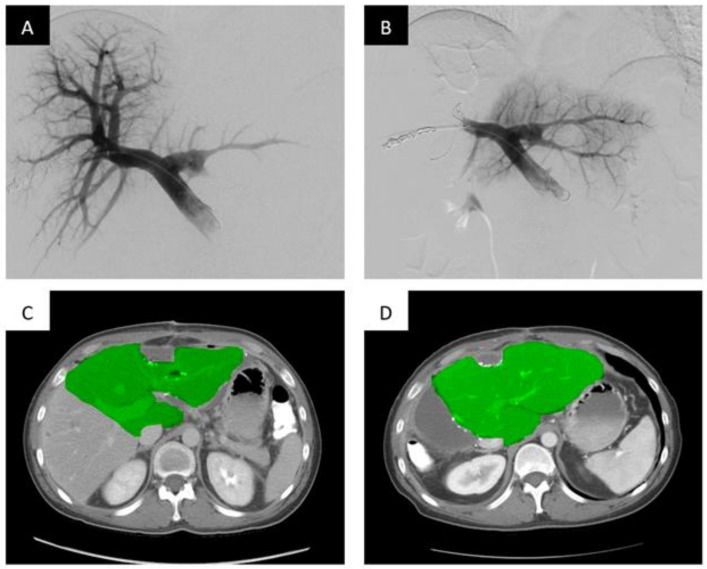
Portal vein embolization of the right portal vein branch with PVA particles and coils demonstrates FLR hypertrophy. Portal vein embolization (PVE) of the right hepatic portal vein with PVA particles and coils. (**A**) Digital subtraction portography prior to embolization via access of the posterior division of the right portal vein. (**B**) Post-embolization single shot image of the right portal system embolized with PVA particles and a Nester coil (Cook Medical, Bloomington, IN, USA). (**C**) Axial contrast enhanced computed tomography (CT) slice demonstrating the pre-embolized liver, with the non-hypertrophied left lobe highlighted. (**D**) Post-PVE contrast-enhanced axial CT slice demonstrating noticeable hypertrophy of the left liver post-right PVE and hepatectomy (approximately 28 days post-resection).

**Figure 2 life-13-00279-f002:**
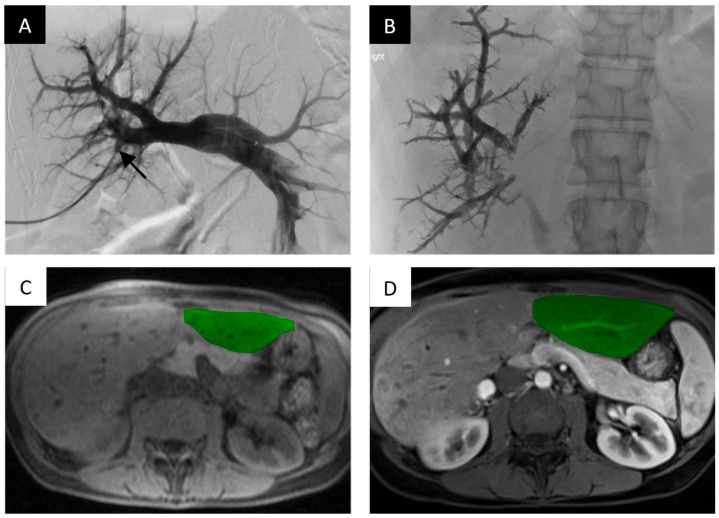
Portal vein embolization of the right portal vein branch with nBCA glue and lipiodol demonstrates FLR hypertrophy. PVE of the right intrahepatic portal vein with nBCA glue with lipiodol. (**A**) Digital subtraction portography via side hole catheter in the main portal vein. Access gained into the portal system via anterior right portal vein branch (black arrow). (**B**) Final image, post-embolization of the right portal vein branches filled with radio-opaque nBCA glue. (**C**) Axial T1-weighted non-contrast magnetic resonance imaging (MRI) slice at the level of the hepatic parenchyma demonstrating the pre-embolized liver, with the non-hypertrophied left lobe highlighted. (**D**) Post-contrast axial T1 weighted MRI slice demonstrating noticeable hypertrophy of the left liver 45 days after embolization.

**Figure 3 life-13-00279-f003:**
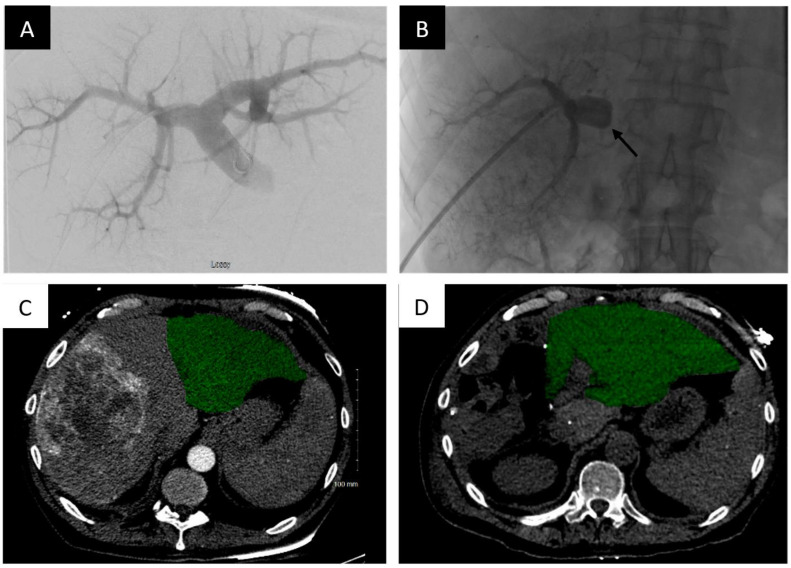
Portal vein embolization of the right portal vein branch with STS foam demonstrates FLR hypertrophy. Portal vein embolization (PVE) of the right hepatic portal vein with STS foam. (**A**) Digital subtraction portography prior to embolization via access of a peripheral right anterior division portal vein branch. (**B**) Post-embolization single shot image of the right portal system opacified with lipiodol-stained STS foam. A 7 F Fogarty balloon (Edwards Lifesciences, Irvine, CA) is used to occlude the proximal right portal vein and maintain patency from the main portal vein to the left portal vein (black arrow). (**C**) Axial contrast-enhanced computed tomography (CT) slice demonstrating the pre-embolized liver, with the non-hypertrophied left lobe highlighted. (**D**) Post-PVE contrast-enhanced axial CT slice demonstrating noticeable hypertrophy of the left liver post-right PVE and hepatectomy (approximately 21 days post embolization).

**Table 1 life-13-00279-t001:** Comparison of the FLR and FLR/TLV% increase for the five embolics described in this review. Increase in FLR% and FLR/TLV of the five accepted embolics for portal vein embolization.

Embolic	Increase FLR (%)	Increase FLR/TLV (%)	Number of Patients
PVA particles			
Camelo et al. [38]	40%	11%	64
Madoff et al. [49]	69%	9.7%	26
Jaberi et al. [50]	n/a	12.3%	40
NBCA glue			
Mukund et al. [39]	52%	14.2%	
Marti et al. [51]	29%	12.5%	52
Ali et al. [52]	49.1%	n/a	52
Luz et al. [53]	52%	12.7%	50
STS foam			
Fischman et al. [42]	n/a	11.9%	35
Marti et al. [51]	25.7%	9.9%	25
Absolute Ethanol			
Sofue et al. [44]	39.9%	12%	83
Santhakumar et al. [54]	43.6%	12.3%	62
Igami et al. [55]	30.0%	11.1%	154
Powdered Gelfoam			
Lainas et al. * [46]	n/a	16.4%	9
Tranchart et al. [56]	29.4%	n/a	20

* Monkey animal model.

**Table 2 life-13-00279-t002:** Comparison of the FLR and FLR/TLV % increase for the experimental portal vein embolization techniques described in this review.

Embolic	Increase FLR (%)	Increase FLR/TLV (%)	Number of Patients
Stem Cells + PVE			
am Esch et al. [83]	+138.66 (mL) *	n/a	11
Furst et al. [84]	77.3%	n/a	6
Branched Chain Amino Acids + PVE			
Beppu et al. [59]	43.8% **	n/a	7
Transarterial chemoembolization followed by PVE			
Ogata et al. [77]	12%	n/a	18
PVE followed by transarterial chemoembolization			
Inaba et al. [78]	1.4x increase ***	n/a	4
Liver Venous Deprivation			
Guiu et al. [71]	40.9%	n/a	7
Guiu et al. [72]	53.4%	n/a	10
PVE + Associating liver partition and portal vein ligation for staged hepatectomy			
Tschuor et al. [61]	41.7%	n/a	3
Ulmer et al. [62]	77.7%	34.9%	9
PVE + Percutaneous intrahepatic split by ablation			
Lunardi et al. [69]	82.9%	n/a	3
Wang et al. [86]	44%	n/a	7

* no % increase reported. Control group (PVE alone) saw an absolute FLR volume increase of 62.95 mL. ** not significant when compared to control group (PVE alone, *p* = 0.112). *** post-PVE + TACE FLR volume compared to pre-PVE + TACE volume.

## Data Availability

Not applicable.

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
