# Peer review of "Portal Vein Embolization: Rationale, Techniques, and Outcomes to Maximize Remnant Liver Hypertrophy with a Focus on Contemporary Strategies"

_life, 2023, doi:10.3390/life13020279_

Round 1
Reviewer 1 Report
I have read with grat interest the paper about PVE, it is an important argument in hepatic surgical oncology. However, several concerns must be addressed before the possibility to publish such a review:
-There are many misleading and misunderstanding definitions, as well as typo errors. "liver insufficiency in the postoperative state" should be replaced throughout the text by post-hepatectomy liver failure (PHLF), since it is an internationally accepted definition. "multiple trials have proved PVE to be safe and effective in down-staging tumoral liver tissue", but PVE is not a down-staging technique, doesn't treat the tumor. On the contrary many authors proposed it can enhance tumoral growth, that is the opposite of down-staging. Please correct. Reference 12 is wrongly cited within the texht, the name of the author is Capussotti. Please check and correct carefully all wrong definitions and typo errors.
-Heading numbers are not correct. Paragraph 2 should be entitled "indications" rather than patients selection, and also contraindications should be reported.
-Authors do not talk about the most important problem of PVE: the failure rate, that is higher than 20%, because of both tumor progressions and inadequate growth of FLR. Please add a small paragraph about PVE failure with risk factors (as you briefely talk of chemotherapy or underlying liver disease etc.; cfr "Soykan EA et al. Predictive Factors for Hypertrophy of the Future Liver Remnant After Portal Vein Embolization: A Systematic Review. Cardiovasc Intervent Radiol. 2021).
-References are not up to date. About PVL, please insert more recent and consistent papers, such as the meta-analysis by Pandanabayana et al, 2015. About PVE, references 13 and 14 are not trials as stated by the authors, do not investigate any dow-staging role, and are not up-to-date. Please correct and replace with newer and larger studies. Regarding data about ALPPS-related morbidity, last results from the ALPPS registry for colorectal liver metastases should be reported since they are more encouraging than for HCC or CCA. Regarding LVD, authors state that only data about 17 patients have been published so fare. Such sentence should be deleted, and the latest references about many refent papers should be added: Kobayashi et al., 2020, Chebaro et al., 2021; Cassese et al 2022 (that interestingly compare LVD to PVE and to ALPPS with long term oncological outcomes).
-In the title the term "pre-operative" before PVE can be removed, and "newer" or other strategies to optimize flr should be emphasized.
-In the abstract the first sentence is not correct, since Liver Transplantation is the only curative treatment, not hepatectomy. Second phrase can be misunderstood, since the indications have not moved from the tumor characteristics (i.e. anatomical resections only for HCC, parenchymal sparing for CRLM, lymphadenectomy for CCA...). Also third sentence should be rephrased without talking about the disease free status, since the hypertrophy techniques are more about avoiding PHLF, that should be cited as a main problem to avoid, while reaching a R0 resection.
-Does Table 2 include only radiologic techniques? Please correct the caption that is not clear, and remove the number of patients or the FLR, since the are referred to only one work rather than all available literature. The difference in the techniques could be added.
Some minor revisions are also required:
-In the INTRODUCTION, when stating " post-resection complications are significantly reduced in patients with an sFLR >20%" please specify only in normal livers. "inability of the non-tumoral tissue" should be replaced with "residual", because is not all the non-tumoral tissue that remains after hepatectomy. Similarly, PVE is more a standard of care than "gold standard".
-About the adequate FLR for cirrhotic and steatotic patients, please, briefely discuss about the existing debate about cutoff (respectively 40-50% and 30-40%). In this light, preoperative functional evaluation can be an additional tool. Reference 24 should be replaced with a more recent one.
-The simple summary should be more incisive. There is no need to write sentences like "The content discussed within this manuscript...Just summarizing directly the arguments."
-About ALPPS, when talking about its possible role, please cite related references (Enne et al.). New minimally invasive technique should also be cited, since it is coherent with future perspective in the argument (mini ALPPS etc.).
-Title of paragraph 5.4 should be "simultaneous" and not "concurrent". Also, please keep referring to this technique as LVD and not HVE, that was sequential and salvage procedure, as described by Hwang in 2009.
-For all the techniques, add briefely some sentences about the oncological outcomes, since they have been big matter of debate for PVE (cfr Giglio et al 2016 for PVE, or Khayat et al 2020 for LVD).
Author Response
Response to Reviewer 1 comments:
“There are many misleading and misunderstanding definitions, as well as typo errors. "liver insufficiency in the postoperative state" should be replaced throughout the text by post-hepatectomy liver failure (PHLF), since it is an internationally accepted definition.”
The appropriate changes to language were made in the manuscript, thank you for the suggestion
"multiple trials have proved PVE to be safe and effective in down-staging tumoral liver tissue", but PVE is not a down-staging technique, doesn't treat the tumor. On the contrary many authors proposed it can enhance tumoral growth, that is the opposite of down-staging. Please correct.”
This language was edited in the manuscript to avoid stating that PVE is effective in down-staging tumoral liver tissue, thank you for the suggestion
“Reference 12 is wrongly cited within the texht, the name of the author is Capussotti. Please check and correct carefully all wrong definitions and typo errors.”
The reference was fixed, thank you for catching the error
-Heading numbers are not correct. Paragraph 2 should be entitled "indications" rather than patients selection, and also contraindications should be reported.
This edit was made in the manuscript, thank you for the suggestion
Authors do not talk about the most important problem of PVE: the failure rate, that is higher than 20%, because of both tumor progressions and inadequate growth of FLR. Please add a small paragraph about PVE failure with risk factors (as you briefely talk of chemotherapy or underlying liver disease etc.; cfr "Soykan EA et al. Predictive Factors for Hypertrophy of the Future Liver Remnant After Portal Vein Embolization: A Systematic Review. Cardiovasc Intervent Radiol. 2021).
A section concerning the PVE failure rate and other potential pitfalls of the procedure was added to the manuscript, thank you for the suggestion
-References are not up to date. About PVL, please insert more recent and consistent papers, such as the meta-analysis by Pandanabayana et al, 2015. About PVE, references 13 and 14 are not trials as stated by the authors, do not investigate any dow-staging role, and are not up-to-date. Please correct and replace with newer and larger studies.
The references were updated to reflect more recent work in the areas of PVL and PVE, and the errors in language regarding the references were fixed, thank you for the suggestions.
Regarding data about ALPPS-related morbidity, last results from the ALPPS registry for colorectal liver metastases should be reported since they are more encouraging than for HCC or CCA. Regarding LVD, authors state that only data about 17 patients have been published so fare. Such sentence should be deleted, and the latest references about many refent papers should be added: Kobayashi et al., 2020, Chebaro et al., 2021; Cassese et al 2022 (that interestingly compare LVD to PVE and to ALPPS with long term oncological outcomes).
Updated information for ALPPS related morbidity was included, as well as updates to the sections on LVD were made, thank you for the suggestions.
-In the title the term "pre-operative" before PVE can be removed, and "newer" or other strategies to optimize flr should be emphasized.
The wording of the title was changed to better reflect the focus of the manuscript, thank you for the suggestion.
-In the abstract the first sentence is not correct, since Liver Transplantation is the only curative treatment, not hepatectomy. Second phrase can be misunderstood, since the indications have not moved from the tumor characteristics (i.e. anatomical resections only for HCC, parenchymal sparing for CRLM, lymphadenectomy for CCA...). Also third sentence should be rephrased without talking about the disease free status, since the hypertrophy techniques are more about avoiding PHLF, that should be cited as a main problem to avoid, while reaching a R0 resection.
The appropriate language changes to the abstract were made, thank you for the suggestions.
-Does Table 2 include only radiologic techniques? Please correct the caption that is not clear, and remove the number of patients or the FLR, since the are referred to only one work rather than all available literature. The difference in the techniques could be added.
Table 2 is highlighting the data of the studies used to highlight the newer strategies that PVE is implementing/being currently researched. It follows the same format as Table 1, and serves to provide a condensed and easy to read version of the data. Thank you for the suggestion.
-In the INTRODUCTION, when stating " post-resection complications are significantly reduced in patients with an sFLR >20%" please specify only in normal livers. "inability of the non-tumoral tissue" should be replaced with "residual", because is not all the non-tumoral tissue that remains after hepatectomy. Similarly, PVE is more a standard of care than "gold standard".
The appropriate language was updated, thank you for the suggestion.
-About the adequate FLR for cirrhotic and steatotic patients, please, briefely discuss about the existing debate about cutoff (respectively 40-50% and 30-40%). In this light, preoperative functional evaluation can be an additional tool. Reference 24 should be replaced with a more recent one.
Added wording to reflect the range of cutoffs, and reference was updated, thank you for the suggestion
-The simple summary should be more incisive. There is no need to write sentences like "The content discussed within this manuscript...Just summarizing directly the arguments."
The wording was updated, thank you for the suggestion.
-About ALPPS, when talking about its possible role, please cite related references (Enne et al.). New minimally invasive technique should also be cited, since it is coherent with future perspective in the argument (mini ALPPS etc.).
The citations and references were updated, as well as the addition of MINI-ALPPS, thank you for the suggestions.
-Title of paragraph 5.4 should be "simultaneous" and not "concurrent". Also, please keep referring to this technique as LVD and not HVE, that was sequential and salvage procedure, as described by Hwang in 2009.
The appropriate language was fixed, thank you for the suggestion
-For all the techniques, add briefely some sentences about the oncological outcomes, since they have been big matter of debate for PVE (cfr Giglio et al 2016 for PVE, or Khayat et al 2020 for LVD).
The appropriate information was updated/sections added, thank you for the suggestion.
Thank you for taking the time to read and review!
Reviewer 2 Report
- The article is devoted to rationale, modern techniques, results, and future directions of preoperative portal vein embolization (PVE). This technique alllows to increase the volume of future liver remnant (ELR) and th decrease the risk of post-resection liver failure. PVE is widely used in clinical practice and both the interventional radiologists and surgeons should be familiar with this technique. The above mentioned review seems to be very useful for readers.
- The review is very well written. The Abstract fully reflects the contents of the article. The quality of references is good ans enough, and will help readers to find additional information. The construction of the paper is logically optimal: successive Patient selection, Methods of PV access, Embolization techniques with analysis of embolic materials and possible complications, and very interesting Future directions. The authors mention dietary supplementation, liver venous deprivation, and stem cells as perspective additions to PVE. The Figures are of very good quality and are clear for sppecialists and non-specialists. I did not find any weakness of this review. In conclusion, in my mind, the paper is very good and for sure will be interesting for readers of your Journal.
Very nice review.
Author Response
Response to Reviewer 2 Comments
The article is devoted to rationale, modern techniques, results, and future directions of preoperative portal vein embolization (PVE). This technique alllows to increase the volume of future liver remnant (ELR) and th decrease the risk of post-resection liver failure. PVE is widely used in clinical practice and both the interventional radiologists and surgeons should be familiar with this technique. The above mentioned review seems to be very useful for readers.
The review is very well written. The Abstract fully reflects the contents of the article. The quality of references is good ans enough, and will help readers to find additional information. The construction of the paper is logically optimal: successive Patient selection, Methods of PV access, Embolization techniques with analysis of embolic materials and possible complications, and very interesting Future directions. The authors mention dietary supplementation, liver venous deprivation, and stem cells as perspective additions to PVE. The Figures are of very good quality and are clear for sppecialists and non-specialists. I did not find any weakness of this review. In conclusion, in my mind, the paper is very good and for sure will be interesting for readers of your Journal.
Very nice review.
Thank you very much for taking the time to read and review this manuscript.
Round 2
Reviewer 1 Report
The overall quality of the manuscript has greatly improved. I think that in this form it can be very interesting for the readers. The abbreviations should be used more consistently (PHLF is presented in subparagraph 1.1 and then again in 1.4; etc.).